# Multidisciplinary Forensic Approach in “*Complex*” Bodies: Systematic Review and Procedural Proposal

**DOI:** 10.3390/diagnostics13020310

**Published:** 2023-01-14

**Authors:** Gennaro Baldino, Cristina Mondello, Daniela Sapienza, Chiara Stassi, Alessio Asmundo, Patrizia Gualniera, Stefano Vanin, Elvira Ventura Spagnolo

**Affiliations:** 1Department of Biomedical and Dental Sciences and Morphofunctional Imaging, University of Messina, Via Consolare Valeria, 1, 98125 Messina, Italy; 2Legal Medicine Section, Department for Health Promotion and Mother-Child Care, University of Palermo, Via del Vespro, 129, 90127 Palermo, Italy; 3Department of Earth, Environmental and Life Sciences (DISTAV), University of Genoa, 16132 Genoa, Italy

**Keywords:** complex cases, decomposed bodies, dismembered bodies, skeletonized remains, charred, forensic pathology, multidisciplinary approach

## Abstract

The recovery of severely altered cadavers (i.e., extensively decomposed, mummified, charred or dismembered) can be a challenge for forensic pathologists due to the difficulties in identification, PMI estimation and manner and cause of death determination. In such cases, integrating routine approaches (autopsy, histology, toxicology) to more specific forensic branches can be fundamental to improving the investigative process. In this paper a systematic review using PubMed, Scopus and Web of Science databases has been performed. The aim was to evaluate the forensic approaches implemented in the management of severely altered bodies due to decomposition, mummification, skeletonization, charring or dismemberment (to which we refer to as “complex”), and the role of each approach in the solution of a case. Then, the literature revision results were used to propose a schematic flowchart summarizing the post mortem activities that can be performed in forensic practice, adaptable in relation to each case.

## 1. Introduction

The evaluation of the cause of death in a forensic context is often challenging, and the autopsy alone may not be conclusive, especially when the corpse is in a severely altered state (decomposed, skeletonized, charred, dismembered, etc.), thus making it difficult to obtain an overview of the features and/or lesions suggestive of the dynamics leading to death [1]. In addition, further issues are associated with the recovery of cadavers from particular sites, such as those buried under building ruins, hidden in secluded places, etc. These conditions can raise challenges not only in terms of victim identification due to the deep alterations produced on the cadaver, but also in terms of acquiring additional information on elucidating the manner of death, the post mortem interval (PMI) and the detection of corpse transfer after the death, especially in cases of suspected homicides. The bodies found in these conditions are defined in this paper as “complex”. In such cases, an exhaustive answer to the investigative questions can only be provided by a proper multidisciplinary approach which correlates the results of the investigations carried out during body recovery site inspection to the circumstantial data and the autopsy findings. Post mortem analyses are further supplemented by toxicological, histopathological and genetic investigations and also, in selected cases, immunohistochemical and imaging techniques (e.g., post mortem CT and MRI) [2,3].

Unfortunately, a specific and validated multidisciplinary approach aimed to achieve a conclusive forensic diagnosis in “complex” cases lacks standardized protocols, despite Interpol protocols/guidelines and the European Council of Legal Medicine (ECLM) suggestions [4,5] which provide partial but useful tools to work in such contexts. Moreover, limits in the application of standardized procedures can be related to the difficulty in routinely using advanced techniques because they are expensive and performed only by specific centers and institutes of forensic medicine [6,7,8].

To this regard, a systematic review of the literature has been carried out. The review aimed to (1) assess the forensic techniques most commonly used by the scientific community in the recovery of “complex” corpses and (2) elaborate a procedural proposal for a complete investigation of these complex cases.

## 2. Materials and Methods

The scientific articles chosen to realize this review were sorted from PubMed (https://pubmed.ncbi.nlm.nih.gov/, accessed on 31 October 2022), Scopus (https://www.scopus.com/home.uri, accessed on 31 October 2022) and Web of Science (https://www.webofscience.com/wos/author/, accessed on 3 January 2023) databases. The review was carried out in accordance with PRISMA guidelines [9]. The inquiry strategy was performed by combining the terms “multidisciplinary approach”, “multidisciplinarity”, or “interdisciplinary approach”, to the keywords “manner of death”, “forensic sciences”, “forensic pathology”, “decomposed bodies”, “skeletonized”, “skeletonization”, “skeletal remains”, “cadaveric dismemberment”, “mummification”, “charred corpses”, “charred bodies”, “burned corpses”, “burned bodies”, or “advanced decomposition” through the Boolean operator “AND”.

Preliminary research led to a total of 692 articles found on PubMed, 539 found on Scopus and 204 found WoS; once the duplicates common to both databases were excluded, the remaining articles were selected according to the following inclusion criteria:(a)Publication date between 1990 and October 2022;(b)Content concerning the management of cases in which “complex” corpses were recovered, namely decomposed, skeletonized, mummified, charred, or dismembered;(c)Cases in which more than one medico-legal investigation was performed, with particular attention to: preliminary inspection performed at the scene of recovery, autopsy, histological, toxicological, radiological, dental, anthropological, entomological and genetic investigations, and additional investigations (i.e., immunohistochemistry, engineering, psychological autopsy, other instrumental investigations).

Bibliographic references from the selected papers were also screened for possible additional inclusions.

This led to the selection of a total of 25 articles [3,6,7,8,10,11,12,13,14,15,16,17,18,19,20,21,22,23,24,25,26,27,28,29,30] (Figure 1) including case reports, case series (referred to as articles discussing ≥ 3 cases) and experimental studies (referred to as articles in which experimental investigations were performed). All selected articles are summarized in Table 1 according to authors, title, year and type of publication.

## 3. Results

All cases were categorized in Table 2 and Table 3 according to the following indicators: number of cases analyzed, personnel heading the survey at the recovery site, site of recovery, state of the corpse upon recovery, circumstantial data, investigations carried out (autopsy, histopathology, radiology, toxicology, entomology, odontology, genetics, anthropology, others), manner and cause of death and post mortem interval (PMI). The main results are presented.

### 3.1. Complex Body Identification

One of the first issues dealing with “complex corpses” is their identification, as not only the time since death, but also several taphonomic factors (e.g., burial, animal scavenging, weathering) can severely alter the body’s physiognomic features. Additional difficulties are associated with dismembered and skeletonized bodies whose remains can be spread in a wide area by the macrofauna activity or the geomorphology of the crime scene. Although DNA profiling is a key element for identification purposes [4], depending on the cadaveric state other approaches can be—or need to be—applied during a forensic investigation. In case of dismembered or skeletonized remains and charred bodies, the identification process starts with the collection of the remains: bone fragments, charred disarticulated remains, victim’s personal objects, etc. Their distinction from not human elements is the next step. Such an issue was managed by Delabarde et al. [10] by associating an anthropologist to the forensic pathologist in a case of recovery of three sets of disarticulated human remains located within a small area of the Amazonian jungle. In such a context, the anthropologic expertise allowed the attribution of the remains to two adult men of Caucasian ancestry, through bone reassembly and exclusion, and subsequent DNA profiling on the remains. Definite identification was provided by the association of these results with the record of missing people from the same area, pinpointing two European tourists. A similar approach was adopted by Baldino et al. [11] in their investigation on a case of an accidental fireworks factory explosion which caused the death of five subjects. While three victims, although severely injured, were recognizable, the other two underwent dismemberment and complete charring, respectively. In the case of the dismembered subject, the lack of an anthropologist made it hard to recover all the scattered remains, so that just a few were collected by the forensic pathologists; DNA extraction from all the remains confirmed that they belonged to the same person, but although the identification could be suggested by the recognition of fragments of clothes on the remains, it was definitely confirmed by comparing the DNA profile to that of the alleged parents. As for the charred body, identification was attributed only based on the evidence on the body of a peculiar necklace, and on the fact that it was referred to be the only woman present at the time of the explosion. Another interesting work is that from Pinchi et al. [12], who engaged in the identification of seven Chinese workers whose charred remains were recovered from a textile factory following a large fire, which exploded due to a malfunction of the heating system. Also in this case, the identification process—particularly challenging also due to the language barrier—required a meticulous combination of the information obtained from the comparison of either the victims’ DNA profile to that of alleged close relatives, odontological investigations, and autopsy findings (e.g., absence of the uterus in a female victim who was referred to have undergone hysterectomy), as well as the recovery of personal objects.

A detailed study of skeletal remains by an expert anthropologist also provides precious information which, once related to circumstantial data, police information and DNA profiling, can contribute to the identification. In this sense, both Manzoni et al. [13] and Jegan et al. [14], with their investigations on mummified, skeletonized and charred bodies, provided complete profiles (ancestry, sex, age and stature of the victims) which restricted the field in the identification process. The anthropological features that were taken into account are: evaluation of skull features (such as the prominence of the zygomatic-maxillary region, the shape of the orbital cavities and the palatine index) for ancestry determination, morphostructural features of pelvic and skull bones for sex determination, and evaluation of the length of long bones (humerus, radius, femur and tibia) for stature estimation. For age estimation, the anthropological evaluation of the degree of fusion of cranial sutures was integrated with the results of the dentition examination performed by a forensic odontologist. The identification process was completed by comparing the victims’ DNA profile to that of the alleged relatives, identified with the help of police investigations which suspected that the corpses could belong to reported missing persons.

The use of forensic odontology can be considered of great value when a dental comparison with either ante mortem clinical records [15,16] or Interpol dental databases [3,4,5] can be performed, in those cases in which no fingerprints are available, or no living relatives exist to compare the DNA profile to. Bublil et al. [7] highlighted the potential of forensic odontology in a case of a female cadaver, decomposed and partially mummified, edentulous and with no living close relatives. Following a thorough inspection of her potential house, a few 10-year-old dental radiographs and a small plastic bag containing isolated teeth were found. The comparison of the ante mortem dental radiographs to X-rays taken from the isolated teeth revealed that they belonged to the same person, based on either the morphology of crowns or roots as well as on the presence of restorative materials in specific surfaces. The DNA extraction from two of the recovered teeth, and its comparison to the DNA of the cadaver showed a complete match, thus allowing the woman’s identification.

### 3.2. Post Mortem Interval Estimation

Next to identification, a major problem when dealing with complex corpses is the PMI estimation. To this regard, due to the impossibility in relying on thanatochronological parameters, a few authors show the value of entomology in forensic investigations for the estimation of long times elapsed from death [17,18,19]. Such essential information can be decisive for the reconstruction of the dynamics of the event, especially in cases of suspected homicide, where the presumed PMI could match with an alleged perpetrator act. An example is provided by Magni et al. [18], whose evaluation of the larval development of Lucilia illustris (Diptera, Calliphoridae), sampled from the cadaver of an 18-year old girl found in a woodland three days after her disappearance, allowed a PMI estimation which revealed concordance with the information provided by the parallel police investigations, thus leading to the imprisonment of the perpetrator.

In their work on the relation between the necrophagous insect community and the human decay pattern carried out on two hanged corpses, Bugelli et al. [17] also shed light on the importance of a correct interpretation of the entomological data, since it strongly depends on several aspects. The authors indeed noted that despite the same manner of death (suicidal hanging) and the same PMI (34 days), the bodies showed two completely different patterns of decomposition, and that the main factors contributing to such differences were identified in the season of recovery, in the geographical location, and in the type of habitat.

It is also worth of mentioning that in very old cases, despite the fact that the time since death cannot be defined, entomological evidence can provide information about the season of the colonization as reported by Vanin et al. [20] and Gaudio et al. [21] working on World War I soldier remains found in the Italian Pre-Alps.

Although hardly considered within forensic investigations, botany represents another valuable approach in complex cases, as reported by Lancia et al. [22] and Caccianiga et al. [23] who were able to determine the PMI of three subjects upon recovery of their skeletal remains by comparing the local vegetation with that found on the recovered bones.

### 3.3. Cause of Death

For the study of injuries suggesting a potential cause of death, an autopsy performed by a forensic pathologist represents the gold standard, especially in cases of complex corpses. In the work by Manzoni et al. [13], the state of the recovered remains was too altered to allow the detection of injuries compatible with gunshot and stab wounds and their paths, and could only be investigated by necroscopic examination. This is also the case reported by Jagen et al. [14], in which the necroscopic evidence of C_2_-C_3_ disarticulation, together with an abnormal mobility at the junction of the greater horn with the body of the hyoid bone and histopathologic evidence of hemorrhagic infiltration of the surrounding soft tissues, suggested a possible strangulation as cause of death, followed by charring and dismemberment. In a second case described by the same authors [14], although the cause of death could not be ascertained, the evidence of a fracture involving the end of the odontoid process with hemorrhagic infiltration of the surrounding soft tissues suggested an ante mortem trauma.

However, when the status of the cadaveric remains is beyond the necroscopic possibility to detect any traumatic injury, forensic radiology—Post Mortem Computed Tomograpy (PMCT) in particular—can be useful for detecting traumatic wounds, fractures and bullets or other retained exogenous material [8,16,24], or to exclude all of them [25,26,27].

Within the context of forensic investigations, although histology often provides unspecific results that need to be co-interpreted with those of other analyses, it can, in some cases, be decisive in the determination of the cause of death. For example, histology was fundamental to define the cause of death of a mummified cadaver as reported by Brett et al. [15]. In this case, once any traumatic lesions were excluded, histological analysis revealed an obstructive calcific coronary arteriosclerosis. The histopathological examination also helped to define the cause of death in the case described by Gitto et al. [25]. The authors of this paper stated that the death was due to an 85% atherosclerotic occlusion of the left anterior descending coronary and an 80% atherosclerotic occlusion of the circumflex. As there was a suspicion of strangulation, the authors further extended the investigations by performing immunohistochemical analyses for the detection of Glycophorin A, an indicator of ante mortem bleeding, on discolored and darker skin areas from the neck. This analysis confirmed the absence of hemorrhage, thus allowing the exclusion of ligature strangulation.

Toxicological investigations also play an important role in the determination of the cause of death, since alcohol, drugs and natural plant poisons (e.g., oleandrin and digoxin) can be fatal—or at least contributory factors to death—in cases of overdose, frequently reflecting a suicidal death [3,28], accidental/recreational assumption, and less frequently poisoning [3,10].

Ventura et al. [26] reported a case in which, in order to determine the cause of death of a female mummified body hidden in a closet by her son, several forensic investigations (on site recovery survey, autopsy, histology, toxicology, genetic analysis) were performed together with a “psychological autopsy”. The latter was performed following the analysis of both subjects’ life conditions and the writings found on their house walls, thus making it possible to define the presence of a complex psychological profile and a perverse mother-child relationship.

### 3.4. Body Dismemberment Methodological Analysis

In cases of dismembered bodies, additional information is provided by PMCT. The act of separating the body parts after death usually raises questions about the origin of the cut wounds: have they been inferred before death, or are they related to the subsequent dismemberment? Are they originated by natural phenomena? In their work, Maiese et al. [8] demonstrated the usefulness of PMCT to answer these questions. These authors, analyzing two cases, showed that the study of the cut surfaces suggested the use of a smooth blade with a penetration and sliding mechanism in association to a cutting and twisting mechanism on the cut surfaces of the upper limbs. In a third case reported by the same authors [8], the study of the cut surfaces and the detection of a blade fragment on PMCT led to the hypothesis that the dismemberment could have been carried out through a serrated saw with a penetration and sliding mechanism. A major precision can be obtained by implementing other approaches to necroscopy and PMCT. For example, the application of stereomicroscopy to the study of bone remains of two of the cases reported by Porta et al. [28] allowed the authors to establish that they were not cut, but totally disarticulated. In a further case, the production of silicon casts from several superficial cut marks detected on the recovered skull, and subsequent analysis by scanning electron microscopy, led to the detection of two patterns of cutting injuries. One pattern corresponded to a weapon with a serrated edge (e.g., a saw), and the second to a weapon with a linear edge (e.g., a common kitchen knife) that were used for dismemberment purposes and to remove the facial soft tissues, respectively. In another case, the authors further applied SEM-EDX, which led to the detection of residues of iron, chromium and nickel (components of cutting weapons) around the margins of the superficial cut lesions, but not in the surrounding environment. With the same aim, Baier et al. [30] relied on micro-CT and 3D methodologies. They recovered two suspicious suitcases which were CT-scanned prior to opening, revealing packed human body parts belonging to a male subject; the second suitcase also contained a saw, a kitchen knife, a hammer and a chisel. Once the victim was identified by DNA profiling, fingerprinting and the evidence of peculiar tattoos, further investigations were carried out at his residence; here, a charred bone fragment was recovered, which was suspected to be the left shoulder joint missing from the contents of the suitcase. A micro-CT examination allowed a virtual extraction of the charred proximal humerus and subsequent alignment with a fragment of left humerus found in one of the suitcases, providing a near perfect geometrical match. The analysis of the cut marks on 3D CT images, compared with the information produced by police investigations, also allowed the identification of the tools used for the dismemberment (an electric carving knife, a saw and a hammer).

## 4. Discussion

If on one side autopsy and routine techniques (histology, toxicology) frequently allow the definition of the types of injuries, and thus the cause of death; on the other side, the determination of the manner of death—homicidal, suicidal or accidental—may require an integration with the information provided by other disciplines. This integration is particularly important and necessary in case of “complex bodies” that represent a challenge in terms of cause and manner of death assessment, PMI estimation and victim identification. Since the issue of forensic analysis in “complex bodies” lacks standardized and shared practical procedures, this review was carried out in order to analyze the cases and procedures reported in the scientific literature which could be useful in elaborating a procedural proposal for complex cases.

Surveys at the site of recovery represent the first approach to the discovery of a cadaver, and since they represent an unrepeatable investigation, the association of a forensic pathologist to the law enforcement officers is of utmost importance in order to correctly collect elements/samples/information useful for the reconstruction of the case [5]. This concept is more than ever valid when managing “complex bodies”, whose state and site of recovery might reflect an attempt of concealment put in place to avoid retrieval, to hide trace evidence and to avoid the identification of the victim [2]. These criminal activities lead very often to a delayed discovery of the corpse which, in turn, is nearly always recovered in an advanced state of putrefaction, skeletonized, or mummified, depending on the environmental conditions of the place where the body was hidden.

Although each case is peculiar and different from another, a preliminary evaluation of the circumstantial data should guide the recruitment of the most adequate forensic experts at the site of recovery/crime scene. For example, in cases of charred or dismembered bodies, especially when the remains are scattered, the survey should be headed not only by the forensic pathologist, but also by a forensic anthropologist and odontologist. The better knowledge of the anthropologist of even highly altered bony remains can help to determine whether any bone or bone fragment is human or not [6].

The presence of a forensic odontologist at the scene of recovery is functional to the recognition—and collection—of eventual teeth or dental fragments which can undergo severe alterations and may thus pass unnoticed.

An entomologist is another expert who should be involved on the body recovery site survey, due to thier usefulness in activities aimed to the (minimum) PMI estimation, such as a proper collection, fixation and subsequent preservation of entomological specimens, and as well as a correct recording of the microclimatic parameters [31,32,33].

In order to determine the cause and manner of death in complex cases, the external examination and autopsy should not be separated from radiological investigations (classic X-rays, PMCT, micro-CT) which provide virtual information for the evaluation, or exclusion, of traumatic injuries in body sites usually not explored during autopsy, or when the corpses are too altered to identify any injury upon inspection [3,7]. PMCT in particular has proved to be an essential tool for the detection of retained material (e.g., bullets), for the study of gunshot/sharp force-related paths, blunt force-related traumas, internal hemorrhages, as well as for the evaluation (together with micro-CT) of minute cut surface characteristics in dismembered bodies in order to identify the weapon used [2,8,11,12,13,34,35,36,37,38,39,40].

To date, autopsy is still considered the gold standard for the determination of the cause of death, especially in cases when no severe cadaveric alterations occur [13,14]. Nonetheless, in some cases of complex corpses, such as extensively decomposed, charred, skeletonized and/or mummified bodies, autopsy findings might be inconclusive. A strong support for understanding the cause of death is provided by histology, although its value is mainly related to the determination of the traumatic or post mortal nature of the injuries in relation to hemorrhagic infiltrations [41,42,43]. Such evidence may not always be easily described when cadavers or cadaveric remains are severely altered or completely charred. For this reason, in complex cases it should be advisable to routinely rely on a more sensitive immunohistochemical approach for the research of markers related to bleeding as signs of a vital reaction [2,8,25,44]. To the best of our knowledge, immunohistochemistry on complex bodies has been performed only by two authors: Maiese et al. [8], who tested IL-15, CD-15 and tryptase on the neck tissues from dismembered and decomposed remains of three subjects, contributing to the strangulation diagnosis in two of them; Gitto et al. [25], who tested glycophorin A on the neck tissues from a mummified cadaver, leading to the exclusion of strangulation as cause of death. As a future perspective, an additional test of the immunohistochemical markers routinely used on “fresher” cadavers, applied on a wider casuistry of complex corpses, could undoubtedly be useful in shedding light on the effective value of immunohistochemistry in highlighting vital reactions as signs of ante and/or peri mortem traumas.

In order to differentiate ante and peri mortem traumas from post mortem skeletal damage, an accurate anthropological analysis can provide useful information, especially when histology cannot detect vital reactions due to severe body conditions (i.e., destructive effect of fire).

Galtés et al. [45] describe the case of a charred body from a fiery car crash, focusing on the different fractural patterns between heat-related and trauma-related fractures. Specifically, bone dehydration due to the exposure to fire is the major determinant for several heat-induced fracture patterns, which include: patina fractures or fine cracks (mainly on the surface of flat bones), longitudinal fractures along the long axis of long bones, curvilinear fractures on long bone shafts, transverse fractures perpendicular to the longitudinal axis of long bones, peeling fractures typical of the epiphyseal regions and step fractures. On the contrary, peri mortem fractures, which occur when the elastic component of the bone still has its water and organic components, show distinctive macroscopic traits, including: horizontal breakage of the cortical layer in the compression side of the diaphysis of long bones, wave lines fractures characterized by a gentle slope and a rapid drop, bone scales occurring close to the margin of the fracture at the compression side, crushed margins (defined as small fractured pieces of bone still attached to the cortical surface at the margin of the compression side of a fracture) and superficial losses of thin pieces of cortical bone (so called “flakes”), which have been related to the presence of surrounding flesh in fresh fractures [40,46].

Even if forensic genetics represents the gold standard for identification purposes, the collaboration between forensic pathologists, anthropologists and odontologists can be crucial. Although a sure identity is provided by the comparison of the DNA profile to that of alleged relatives, several cases exist in which, due to the lack of close relatives, no comparative genetic analyses can be performed. In such a context, an integrated approach with anthropological and dental evaluations can be conclusive [7,8]. In skeletonized bodies, studying the remains can provide precious information about the age, sex, race and stature of the victim, as well as skeletal changes characteristic of specific diseases [6]. Moreover, in this setting, a useful contribution can be provided by an odontologist: if ante mortem dental records of the alleged decedent are available, the cadaveric identification can be provided by a comparison to previous odontostomatological pictures of the person. When no such information is available, examination of the dentition can provide an estimation of the age of the decedent [47,48,49]. In addition, analysis of maxillofacial structures can lead to the identification of distinguishing marks, losses of substance and/or fractures [2,3,6]. Other important tools for positive personal identification are implanted medical devices, such as orthopaedic metallic implants, vascular stents, vascular occluding devices, prosthetic heart valves, ventricular assist devices, implantable cardiac defibrillators and pacemakers [50,51]. Particularly, the usefulness of orthopaedic devices for forensic identification is well described using the device serial number, patient-specific database, manufacturer information (i.e., shipping details, details of the distributors, details on the hospitals that have the devices) and clinical records [52,53,54].

Thus, anthropological and dental analyses are important in the management of complex cases, due to their contribution to the identification process in association with eventual DNA profiles, fingerprint analyses and results from parallel investigations. Such approaches aren’t always globally considered, and frequently identification relies on circumstantial data or on the recognition of clothes/personal effects worn by the victim. This could easily mislead the investigations, especially in cases of suspected homicide when garments/objects/clothes are supposedly placed on the victim [55]. For this reason, a DNA profile—whenever applicable—should always be provided [56].

Moreover, the activities of a forensic pathologist should be integrated with the work of entomologists for providing a chronological gold standard for the PMI estimation in severely altered corpses [31,32,33,57,58], based on the detailed analysis and study of the insects correlated with the crime scene environment and microclimatic conditions. The analysis of the developmental status of the larvae of flies, especially in the family Calliphoridae, Sarcophagidae and Muscidae, which are among the first colonizer of a corpse, enable the estimation of the minimum post mortem interval (min PMI) if the temperatures before the body recovery are known. Such an approach can be applied during the first wave of colonization, and therefore to intervals of a few weeks, for average temperatures. In such a context, compared to the evaluation of the external morphological characteristics alone, a higher degree of accuracy has been recently provided by micro-CT, for its capability to precisely estimate the age of blowfly pupae by a combination of external and internal morphological characteristics [59]. For longer PMIs, it is essential to study the communities that follow one another on the corpse, each linked to particular states of decomposition of the body. Moreover, entomology can also elucidate whether the decomposition process really took place at the site of recovery, or if the corpse has been moved after death [60,61,62]. It is worth of mentioning that the entomological approach was shown to be more accurate than the Total Body Score methods for time since death estimation as published by Franceschetti et al. [63].

Among the most recent approaches introduced in forensic investigations, it is worth mentioning the so called “psychological autopsy”, a term coined by Schneidman who described it as “nothing less than a thorough retrospective investigation of the intention of the deceased” [26]. The methodology of such approach relies on the collection of any source of information, including death scene examinations, police reports, witness statements and documents related to life circumstances (school records, medical records, work information, etc.) [64,65].

In conclusion, forensic investigations can rely on several approaches to reconstruct complex cases, each one representing a piece of a “puzzle”. To this regard, the analysis of the reviewed articles confirms not only the importance of the presence of a forensic examiner at the site of recovery, but in selected cases, also, the need of a multidisciplinary approach to shed light on the dynamics of the events, to correctly identify the bodies recovered and to estimate the post mortem interval. In such a context, anthropological, odontological, genetic, radiologic and entomological investigations can play a leading role, especially when facing complex cases. The multidisciplinary approach can also be supported in selected cases by additional investigations such as forensic botany, a recent discipline not yet widely used for judicial purposes, which nonetheless can prove to be useful for several purposes such as the determination of the location of a burial and the interval since deposition [65,66,67,68]. Other investigations include forensic veterinary sciences, genetic investigations on animals, forensic geology, psychological autopsy, engineering, etc.

Thus, on the basis of the evidence emerging from the present literature review, a schematic procedural flowchart for the management of complex bodies has been proposed in Figure 2, aiming to provide an overview of (1) the experts who could give a useful contribution to case analyses and, also, (2) the investigations which should be performed, whenever available, to help solve the issues related to the cause and manner of death, victim identification and post mortem interval estimation.

## Figures and Tables

**Figure 1 diagnostics-13-00310-f001:**
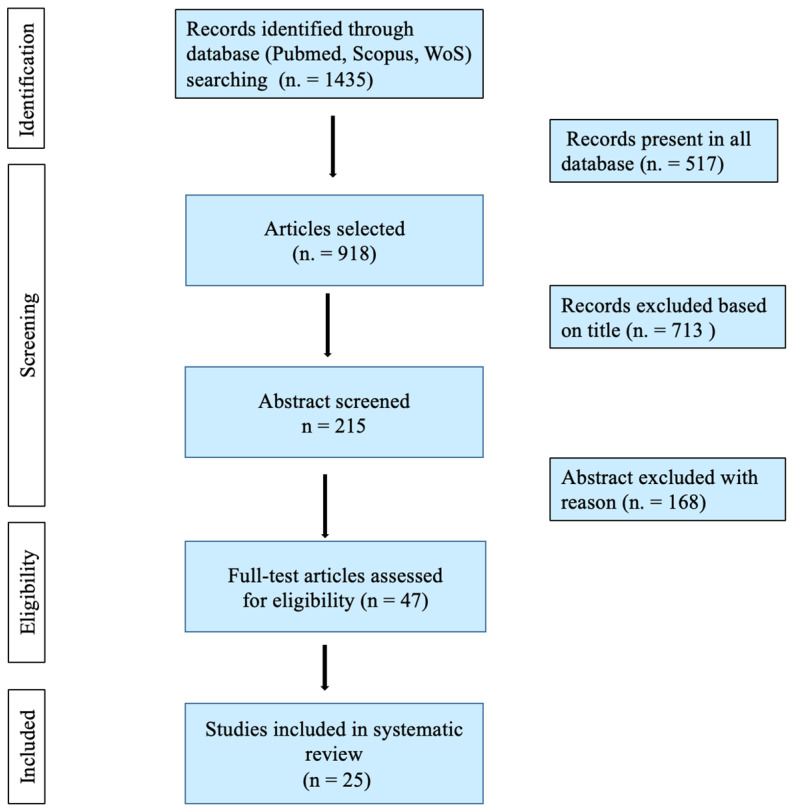
Flowchart of the article selection process.

**Figure 2 diagnostics-13-00310-f002:**
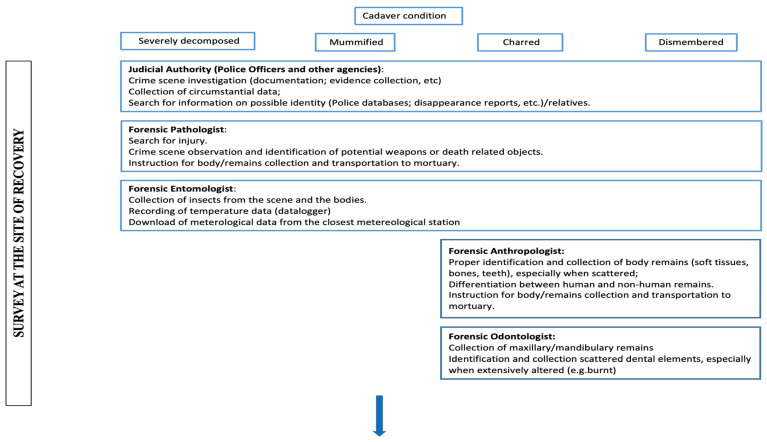
Procedural flowchart proposed.

**Table 1 diagnostics-13-00310-t001:** Articles included in the revision; first author, reference, title, publication year and article type were schematically reported.

Author	Title	Year	Study Design
Harding et al. [3]	Multidisciplinary Investigation of an Unusual Apparent Homicide/Suicide	2008	Case report
Archer et al. [6]	Social Isolation and Delayed Discovery of Bodies in Houses: The Value of Forensic Pathology, Anthropology, Odontology and Entomology in the Medico-Legal Investigation	2005	Case report
Bublil et al. [7]	Antemortem and Postmortem Nonapposite Data—A Multidisciplinary Identification Strategy	2015	Case report
Maiese et al. [8]	Corpse Dismemberment: A Case Series. Solving the Puzzle through an Integrated Multidisciplinary Approach	2020	Case series
Delabarde et al. [10]	Missing in Amazonian Jungle: A Case Report of Skeletal Evidence for Dismemberment	2010	Experimental study
Baldino et al. [11]	Forensic Investigative Issues in a Fireworks Production Factory Explosion	2021	Case series
Pinchi et al. [12]	Deaths Caused by a Fire in a Factory: Identification and Investigative Issues	2016	Case report
Manzoni et al. [13]	Discovering a Double Murder through Skeletal Remains: A Case Report	2018	Case report
Jegan et al. [14]	When Skeletons Tell their Tales: 2 Instances of Examination of Skeletonised Human Remains. A Case Report	2020	Case report
Brett et al. [15]	The Phenomenon of the Urban Mummy	2015	Case report
de Boer et al. [16]	Disaster Victim Identification Operations with Fragmented, Burnt, or Commingled Remains: Experience-Based Recommendations	2020	Case series
Bugelli et al. [17]	Decomposition Pattern and Insect Colonization in Two Cases of Suicide by Hanging	2018	Experimental study
Magni et al. [18]	Entomological Evidence: Lesson to Be Learnt from a Cold Case Review	2012	Case report
Garcìa et al. [19]	The Paradigm of Interdisciplinarity in Forensic Investigation. A Case in Southeastern Spain	2021	Case report
Vanin et al. [20]	Forensic Entomology and Archaeology of War	2010	Case report
Gaudio et al. [21]	Excavation and Study of Skeletal Remains from a World War I Mass Grave	2013	Case series
Lancia et al. [22]	The Use of Leptodyctium riparium (Hedw.) Warnst in the Estimation of Minimum Postmortem Interval	2013	Case report
Caccianiga et al. [23]	Common and much Less Common Scenarios in which Botany is Crucial for Forensic Pathologist and Anthropologists: A Series of Eight Case Studies	2021	Case series
Pomara et al. [24]	A Multidisciplinary Approach to the Investigation of a Collapsed Building	2010	Case series
Gitto et al. [25]	A Scream from the Past: A Multidisciplinary Approach in a Concealment of a Corpse Found Mummified	2015	Case report
Ventura et al. [26]	The Need for an Interdisciplinary Approach in Forensic Sciences: Perspectives from a Peculiar Case of Mummification	2013	Case report
Cortellini et al. [27]	Living with the Dead: A Case Report and Review of the Literature	2019	Case report
Palazzo et al. [28]	Integrated Multidisciplinary Approach in a Case of Occupation Related Planned Complex Suicide-Peticide	2020	Case report
Porta et al. [29]	Dismemberment and Disarticulation: A Forensic Anthropological Approach	2016	Case series
Baier et al. [30]	Novel Application of Three-Dimensional Technologies in a Case of Dismemberment	2017	Case report

**Table 2 diagnostics-13-00310-t002:** Summary of the data about each case of “complex” body analyzed in the selected articles. (M: Male; F: Female; FP: Forensic Pathologist; NA: Not Applicable; ND: Not Determined; PO: Police Office).

Ref.	Sex	Age	Survey	Circumstantial Data	Recovery Site	Cadaver Status	Cause of Death	Manner of Death	PMI
[3]	F	63	PO–FP	Yes	Home	Mummified; skeletonized	Poisoning	Suicide	~3 y
	M	34	PO–FP	Yes	Home	Mummified; skeletonized	Poisoning	Homicide	~3 y
	Dog	-	PO–FP	Yes	Home	Mummified; skeletonized	Poisoning	Peticide	~3 y
[6]	F	75	PO–FP	Social isolation	Home	Skeletonized	Cardiac failure	Natural causes	~18 m
	M	73	PO–FP	Social isolation	Home	Skeletonized	Cardiac failure	Natural causes	7–10 m
[7]	F	ND	PO–FP	Social isolation	Home	Putrefied; partially mummified	Cardiac failure	Natural causes	ND
[8]	F	49	FP	NA	Edge of rural road	Dismembered and putrefied remains inside multiple bags	Strangulation	Homicide	ND
	F	59	FP	NA	Dustbins	Dismembered and putrefied remains inside two bags	Strangulation	Homicide	ND
	F	18	FP	NA	Edge of country road	Dismembered and putrefied remains inside two luggages	Abdomen stab	Homicide	ND
[10]	M	ND	PO	Accident during a trip in the woods	Amazonian jungle	Dismembered remains, partly skeletonized	Hallucinogenic poisoning	Accidental	~4 m
	M	ND	PO	Accident during a trip in the woods	Amazonian jungle	Dismembered remains, partly skeletonized	Hallucinogenic poisoning	Accidental	~4 m
[11]	M	34	PO–FP	Explosion during welding activity of the factory’s gates	Remains spread in the surroundings of fireworks factory	Dismembered remains	Explosion-related dismemberment	Accidental	<24 h
	F	71	PO–FP	Explosion during welding activity of the factory’s gates	Surroundings of fireworks factory	Completely charred	Fire-related charring	Accidental	<24 h
	M	39	PO–FP	Explosion during welding activity of the factory’s gates	Surroundings of fireworks factory	Advanced carbonization: missing upper and lower limbs	Fire-related charring	Accidental	<24 h
	M	71	PO–FP	Explosion during welding activity of the factory’s gates	Surroundings of fireworks factory	Advanced carbonization: missing upper and lower limbs	Fire-related charring	Accidental	<24 h
	M	23	PO–FP	Explosion during welding activity of the factory’s gates	Surroundings of fireworks factory	Completely charred	Fire-related charring	Accidental	<24 h
[12]	M	ND	PO–FP	Explosion during welding activity in a textile factory	In a textile factory	Advanced carbonitaion	Poisoning	Accidental	<24 h
	M	ND	PO–FP	Explosion during welding activity in a textile factory	In a textile factory	Completely charred	Poisoning	Accidental	<24 h
	M	ND	PO–FP	Explosion during welding activity in a textile factory	In a textile factory	Advanced carbonization, lower limb missing	Poisoning	Accidental	<24 h
	M	ND	PO–FP	Explosion during welding activity in a textile factory	In a textile factory	Advanced carbonization	Poisoning	Accidental	<24 h
	F	ND		Explosion during welding activity in a textile factory	In a textile factory	Dismembered remains	Poisoning	Accidental	<24 h
	F	ND	PO–FP	Explosion during welding activity in a textile factory	In a textile factory	Advanced carbonization, missing upper and lower limbs, exposure of the abdominal organs	Poisoning	Accidental	<24 h
	M	ND		Explosion during welding activity in a textile factory	In a textile factory	Advanced carbonization: missing upper and lower limbs	Poisoning	Accidental	<24 h
[13]	M	ND	FP	NA	Woods	Skeletonized; partially mummified	Sharp force and gunshot-related thoracic trauma	Homicide	2–3 m
	M	ND	FP	NA	Woods	Skeletonized	Gunshot-related polytrauma	Homicide	2–3 m
[14]	M	45–50	PO–FP	NA	Outdoor	Dismembered; charred remains	Supposed strangulation	Supposed homicide	~3 m
	M	~30		NA	Rise farm	Skeletonized remains	ND	ND	~3 m
[15]	M	59	PO	Couldn’t be reached by his sister; history of alcoholism	Home	Mummified	Atherosclerotic heart disease	Natural causes	~18 m
	M	33	PO	Broke up with girlfriend	Landfill	Mummified	Hanging-related asphyxia	Suicide	~2 m
[16]	>3000 M & F	0	PO	Terroristic attack	Twin Towers	Fragmented/decomposed/burnt/commingled remains	Compound	Homicide	NA
				Air crash	-	Dismembered remains with thermal injuries	Compound	Accidental	NA
				Bushfires	-	Fragmented and burnt remains	Fire-related	Accidental	NA
				Terroristic attack	-	Fragmented/decomposed/burnt/skeletonized remains	Compound	Homicide	NA
				Air crash	-	Dismembered/commingled/decomposed remains	Compound	Accidental	NA
				Terroristic attack	-	Dismembered bodies	Compound	Homicide	NA
				Fire in apartment block	-	Fragmented and charred bodies/remains	Compound	Accidental	NA
[17]	M	24	PO–FP		Country	Partially mummified	Hanging	Suicide	34 d
	M	15	PO–FP		Woods	Skeletonized	Hanging	Suicide	34 d
[18]	F	18	PO	NA	Woods	Putrefied	Blunt force-related head trauma	Homicide	3 d
[19]	M	ND	PO	NA	Countryside	Mummified	Cardiac failure	Natural causes	~15 m
[20]	M	16–19	PO-FP	ND	Buried under 0.60 m of soil in a mountain	Skeletonized remains	Traumatic head injuriesfollowing a grenade explosion	ND	ND
[21]	M	18–23	PO-FP	Body remains from World War	World war I Mass Grave on the mountains	Skeletonized	ND	ND	ND
	M	23–29	PO-FP	Body remains from World War	World war I Mass Grave on the mountains	Skeletonized			
	M	26–34	PO-FP	Body remains from World War	World war I Mass Grave on the mountains	Skeletonized			
	M	18–23	PO-FP	Body remains from World War	World war I Mass Grave on the mountains	Skeletonized			
	M	18–23	PO-FP	Body remains from World War	World war I Mass Grave on the mountains	Skeletonized			
	M	20–34	PO-FP	Body remains from World War	World war I Mass Grave on the mountains	Skeletonized			
	M	26–35	PO-FP	Body remains from World War	World war I Mass Grave on the mountains	Skeletonized			
[22]	F	ND	JA-FP	ND	Wooded area	Skeletonized remains	ND	ND	--
	F	ND	PO-FP	Disappeared	Overgrown banks of a river, entangled in a bush	Skeletonized remains	ND	Homicide	~3 m
	F	ND	PO-FP	Disappeared	Woodland area	Skeletonized remains	Skull traumatic injury by a blunt object	Homicide	1 y
	M	ND	PO-FP	ND	Uncultivated areaadjacent to a main road	Skeletonized remains	ND	ND	>10 y
	M	ND	PO-FP	ND	On a river bank	Skeletonized remains	ND	ND	~15 y
[23]	F	ND	PO-FP	Disappeared	Field in a sparsely populated industrial area	Putrefied	ND	Homicide	~3 m
	F	ND	PO-FP	Disappeared	Buried in a shallow grave in a suspect’s orchard.	Putrefied and dismembered	ND	Homicide	1 m
	M	ND	PO-FP	Disappeared	Plastic sack in the tanks of a hydroelectric power station	Skeletonized and charred	ND	Homicide	<20 m
	ND	ND	PO-FP	ND	Semi-closed suitcase in a wood land area	Partially skeletonized human remains	ND	ND	~6 m
[24]	8 M & F	-	PO–FP	Explosion of domestic gas cylinder	Buried under ruins	Severely injured; dismembered	Traumatic asphyxia (n = 6); Severe injuries (n = 2)	Accidental	<24 h
[25]	M	83	PO	Hidden by the son to obtain pension	Inside a bag, walled	Mummified	Cardiac failure	Natural causes	2 y
[26]	F	ND	PO–FP	-	Wardrobe	Mummified	Cardiac failure	Natural causes	~3 y
[27]	M	ND	PO	-	Home	Mummified; partially skeletonized	Cardiac failure	Natural causes	~2 m
[28]	F	60	PO	Depression	Home	Putrefied	Hanging	Suicide	~30 d
	Dog	-	PO	-	Home	Putrefied	Poisoning	Peticide	ND
[29]	F	77	PO	-	Home	Dismembered; putrefied remains	Blunt force-related head trauma	Homicide	ND
	?	-	PO	NA	Cage	Dismembered	Sharp force-related injuries	ND	ND
	M	-	PO	NA	Woods	Dismembered	Sharp force-related injuries	Homicide	ND
	F	28–43	PO	NA	Submerged	Dismembered; adipocere	Sharp force-related injuries	Homicide	ND
	F	38–52	PO	NA	Woods	Dismembered	Sharp force-related injuries	Homicide	ND
	M	76	PO	-	Farmhouse	Dismembered	Blunt force-related head trauma and stabbing	Homicide	<24 h
[30]	M	ND	PO	-	River bank	Dismembered; putrefied remains	ND	Homicide	ND

**Table 3 diagnostics-13-00310-t003:** Schematic representation of the investigations performed in each case reported in the reviewed literature (PMCT: post mortem computed tomography; IHC: immunohistochemistry; IL: interleukin; HbCO: carboxyhemoglobin; EDDP: 2-ethylidene-1, 5-dimethyl-3, 3-diphenylpyrrolidine).

Ref.	Autopsy	Radiology	Histology	Toxicology	Entomology	Odontology	Genetics	Anthropology	Other
[3]	Yes	X-rays	No	Positive for benzodiazepines	No	Comparison of ante mortem and post mortem dentition	No	Yes	No
	Yes	X-rays	No	Positive for benzodiazepines	No	Comparison of ante mortem and post mortem dentition	No	Yes	No
	Yes	X-rays	No	No	No	No	No	No	Forensic veterinary
[6]	Yes	No	No	No	Yes	Yes	Yes	Yes: bony alterations consistent with Paget’s disease	No
	Yes	No	No	No	Yes	Yes	No	Yes	No
[7]	Yes	X-rays	No	No	No	Comparison of ante mortem and post mortem dentition	Yes	No	Finger-prints
[8]	Yes	PMCT	Yes	Alcohol; morphine; methadone; EDDP	No	No	Yes	No	IHC: IL15; CD15; tryptase
	Yes	PMCT: face and neck fractures	Yes	Yes: negative	No	No	Yes	No	
	Yes	PMCT: right fronto-parietal haematoma	Yes	Morphine; codeine	No	No	Yes	No	
[10]	Yes	No	No	No	No	Yes	Yes	Yes	Comparison with studies on pigs for the determination of the cutting tool
[11]	No	No	Yes	No	No	Yes	Yes	No	Engineer
	Yes	3D PMCT	Yes	HbCO < 10%	No	No	No	No	
	Yes	3D PMCT	Yes	HbCO < 10%	No	No	No	No	
	Yes	3D PMCT	Yes	HbCO 82%	No	Yes	No	No	
	Yes	3D PMCT	Yes	HbCO < 10%	No	No	Yes	No	
[12]	Yes	X-rays	No	HbCO% 48.68%; cyanides 8.85 mcg/mL	No	Comparison of ante mortem and post mortem dentition	Yes	No	Finger-prints
	Yes	X-rays	No	HbCO% 61.63%; cyanides 5.17 mcg/mL	No	Comparison of ante mortem and post mortem dentition	Yes	No	No
	Yes	X-rays	No	HbCO% 81.23%; cyanides 3.63 mcg/mL	No	Comparison of ante mortem and post mortem dentition	Yes	No	No
	Yes	X-rays	No	HbCO% 95.77%; cyanides 1.25 mcg/mL	No	Comparison of ante mortem and post mortem dentition	Yes	No	Finger-prints
	Yes	X-rays	No	HbCO% 94.4% and cyanides 0.93 mcg/mL	No	No	Yes	No	No
	Yes	X-rays	No	Positive for HbCO% 94.89% and cyanides 2.59 mcg/mL	No	No	Yes	No	No
	Yes	X-rays	No	Positive for HbCO% 88.05% and cyanides 0.29 mcg/mL	No	No	Yes	No	No
[13]	Yes	X-rays: multiple thoracic traumas	Yes	No	No	NP	Yes	Yes	No
	Yes	X-rays: multiple face, neck and thoracic traumas	Yes	No	No	NP	Yes	Yes	No
[14]	Yes	No	Yes	No	No	Yes	No	Yes	No
	NP	No	No	No	No	Yes	No	Yes	No
[15]	Yes	No	Calcific coronary arteriosclerosis	No	No	Comparison of ante mortem and post mortem dentition	No	NP	No
	Yes	No	No	No	No	No	No	No	No
[16]	No	X-rays	No	NP	No	Comparison of ante mortem and post mortem dentition	Yes	Yes	No
	No	No	No	No	No	NP	NP	Yes	Engineer
	No	X-rays	No	No	No	Comparison of ante mortem and post mortem dentition	Yes	Yes	No
	No	PCT	No	No	No	Comparison of ante mortem and post mortem dentition	Yes	Yes	Finger-prints
[17]	Yes	PMCT	No	No	Yes	No	No	No	Botany; meteorology
	Yes	PMCT	No	No	Yes	No	No	No	Botany; meteorology
[18]	Yes	No	No	No	Yes	No	Yes	No	No
[19]	Yes	No	No	No	Yes	No	Yes	No	Fingerprints
[20]	No	No	No	No	Yes	Yes	No	Yes	No
[21]	No	No	No	No	No	Yes:	No	Yes	3D Scan
	No	No	No	No	No	Yes:	No	Yes	
	No	No	No	No	No	Yes:	No	Yes	
	No	No	No	No	No	Yes:	No	Yes	
	No	No	No	No	No	Yes:	No	Yes	
	No	No	No	No	No	Yes:	No	Yes	
	No	No	No	No	No	Yes:	No	Yes	
[22]	No	X-rays	No	No	No	Yes:	Yes	Yes	Botany
	Yes	No	No	No	No	No	Yes	No	Botany, Archeology
	No	No	No	No	No	No	Yes	Yes	Botany
	Yes	No	No	No	No	No	Yes	Yes	Botany
	No	No	No	No	No	Yes	No	Yes	Archeology, Botany
[23]	Yes	No	No	No	No	No	Yes	No	Botany, Archeology
	Yes	No	No	No	No	No	Yes	No	Botany
	No	No	No	No	No	No	Yes	No	Botany
	No	No	No	No	No	No	No	Yes	Botany
[24]	Yes	X-rays	Yes	No	Yes	No	No	No	Engineer
[25]	Yes	PMCT	Yes	Negative	No	No	Yes	No	IHC: glycophorin A
[26]	Yes	X-rays	No	Yes: negative	No	No	Yes	No	Psychological autopsy
[27]	Yes	X-rays	No	No	No	No	No	No	No
[28]	Yes	No	No	Clotiapine; phenobarbital	No	No	No	No	No
	Yes	No	No	No	No	No	No	NP	Veterinary
[29]	Yes	No	No	No	No	No	No	Yes	Stereomicroscopy; casts
	No	No	No	No	No	No	No	Yes	
	Yes	No	No	No	No	No	No	Yes	
	NP	No	No	No	No	No	No	Yes	Stereo-micro-scopy; SEM-EDX; casts; veterinary
	NP	No	No	No	No	No	No	Yes	
	Yes	No	No	No	No	No	No	Yes	
[30]	Yes	PMCT	No	No	No	NP	Yes	Yes	Micro-CT; 3D print

## Data Availability

Not applicable.

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
