# Peer review of "Multidisciplinary Forensic Approach in “Complex” Bodies: Systematic Review and Procedural Proposal"

_diagnostics, 2023, doi:10.3390/diagnostics13020310_

Round 1

Reviewer 1 Report

This is sure a valuable contribution, but I have certain misgivings in regard the literature review methodology. DOI: 10.1096/fj.07-9492LSF might help in future reviews. Honestly, I’d like to see this published; it’s just that I consider Scopus and Pubmed too similar. 

1. I believe the “Abstract” section should be:a single paragraph. Abstracts should give a pertinent overview of the work. Use of the outlay style of structured abstracts, but without headings makes sense only in research articles. The abstract should be an objective representation of the article and should not exaggerate the main conclusions.

2. Lns 72, 73&74 – literature search strategy – in 2006 Burnham concluded that Scopus and WOS complement each other as neither resource is all-inclusive, so m not quite convinced in your literature search.  Even though authors declare their adherence to the PRISMA guidelines, tvey refere to the paper of Page et al. from 2021 (ref. 9 - Page, M.J.; McKenzie, J.E.; Bossuyt, P.M.; Boutron, I.; Hoffmann, T.C.; Mulrow, C.D.; Shamseer, L.; Tetzlaff, J.M.; Akl, E.A.; 476 Brennan, S.E.; Chou, R.; Glanville, J.; Grimshaw, J.M.; Hróbjartsson, A.; Lalu, M.M.; Li, T.; Loder, E.W.; Mayo-Wilson, E.; 477 McDonald, S.; McGuinness, L.A.; Stewart, L.A.; Thomas, J.; Tricco, A.C.; Welch, V.A.; Whiting, P.; Moher, D. The PRISMA 2020 478 Statement: An Updated Guideline for Reporting Systematic Reviews. BMJ 2021, 372, n71.). It is extremely unlikely that these authors are content with scoping these two databases.  Besides, PRISMA guidelines suggest including reports of included studies, what was not the case.

3. Total number of references is totally too modest.

4. Procedural flowchart proposed in figure 2. is confusing and extremely hard to follow. You have way too much text in your graphics – that is the place for short notes rather than full sentences. Besides, it should be clear, lucid and SELFEXLANATORY.

5. Instead of revising the forensic entomology based on ancient works that are mostly outdated (really, you quoted Amendt, J.; Campobasso, C.P.; Gaudry, E.; Reiter, C.; LeBlanc, H.N.; Hall, M.J. European Association for Forensic Entomology. 526 Best Practice in Forensic Entomology -- Standards and Guidelines. Int J Legal Med 2007, 121(2), 90–104) as examples of “best practice” – you should examine the newest Bharti, Meenakshi, and Devinder Singh. "Insects in Forensic Investigations." Insects as Service Providers. Springer, Singapore, 2023. 165-182.

6. The “Last but not least” phrase from the ln 395 is not the part of a standard scientific style.

Author Response

REVIEWER 1

Dear Reviewer,

 thank you for your comment; the suggested changes have been performed and you can find the manuscript modifications as follows: the added text in red; the deleted text in blue. Tables and figures have been modified.

This is sure a valuable contribution, but I have certain misgivings in regard the literature review methodology. DOI: 10.1096/fj.07-9492LSF might help in future reviews. Honestly, I’d like to see this published; it’s just that I consider Scopus and Pubmed too similar. 

  1. I believe the “Abstract” section should be:a single paragraph. Abstracts should give a pertinent overview of the work. Use of the outlay style of structured abstracts, but without headings makes sense only in research articles. The abstract should be an objective representation of the article and should not exaggerate the main conclusions.

Thank you for your suggestion. The abstract was structured following the author’s instructions; however, we agree with your suggestion and the abstract has been modified

  1. Lns 72, 73&74 – literature search strategy – in 2006 Burnham concluded that Scopus and WOS complement each other as neither resource is all-inclusive, so m not quite convinced in your literature search.  Even though authors declare their adherence to the PRISMA guidelines, tvey refere to the paper of Page et al. from 2021 (ref. 9 - Page, M.J.; McKenzie, J.E.; Bossuyt, P.M.; Boutron, I.; Hoffmann, T.C.; Mulrow, C.D.; Shamseer, L.; Tetzlaff, J.M.; Akl, E.A.; 476 Brennan, S.E.; Chou, R.; Glanville, J.; Grimshaw, J.M.; Hróbjartsson, A.; Lalu, M.M.; Li, T.; Loder, E.W.; Mayo-Wilson, E.; 477 McDonald, S.; McGuinness, L.A.; Stewart, L.A.; Thomas, J.; Tricco, A.C.; Welch, V.A.; Whiting, P.; Moher, D. The PRISMA 2020 478 Statement: An Updated Guideline for Reporting Systematic Reviews. BMJ 2021, 372, n71.). It is extremely unlikely that these authors are content with scoping these two databases.  Besides, PRISMA guidelines suggest including reports of included studies, what was not the case. 

Thank you for your suggestions. We have extended the review using also Web of Science database. Thus, Figure 1 has been modified. The analysis of the new database didn’t provide other articles to enroll in the review.

  1. Total number of references is totally too modest.

Thank you for the suggestion. Other references have been added in the bibliography.

  1. Procedural flowchart proposed in figure 2. is confusing and extremely hard to follow. You have way too much text in your graphics – that is the place for short notes rather than full sentences. Besides, it should be clear, lucid and SELFEXLANATORY. 

Thank you for your suggestion. We have modified the flowchart as suggested.

  1. Instead of revising the forensic entomology based on ancient works that are mostly outdated (really, you quoted Amendt, J.; Campobasso, C.P.; Gaudry, E.; Reiter, C.; LeBlanc, H.N.; Hall, M.J. European Association for Forensic Entomology. 526 Best Practice in Forensic Entomology -- Standards and Guidelines. Int J Legal Med 2007, 121(2), 90–104) as examples of “best practice” – you should examine the newest Bharti, Meenakshi, and Devinder Singh. "Insects in Forensic Investigations." Insects as Service Providers. Springer, Singapore, 2023. 165-182.

Thank you for your suggestion. The citation of the suggested chapter was added to the ms.  Because the ms was prepared during the 2022 we were not aware of this publication.

  1. The “Last but not least” phrase from the ln 395 is not the part of a standard scientific style.

Thank you. The change has been provided.

Reviewer 2 Report

This manuscript presents the results of investigations on a relevant subject matter of Journal «Diagnostics». The manuscript presents an analysis of approaches to forensic examination of bodies. In my opinion, this is an extremely important problem in forensic medicine, but the manuscript needs to be improved.

Comments

1). English cannot be considered as a criterion, since murder is a crime against humanity. It is necessary to use all available references so as not to miss important information. In addition, excellent text translators are available on the Internet.

2). Tables 1 and 2: Remove table break in the middle of the page (Table 1(2). (Continued)).

3). The title of section 3.1 and 3.3 are the same. Change the titles to indicate the specific content of the section.

4). Of course, DNA analysis is an important part of personality identification, but on its basis it is impossible to establish the cause and date of death. Therefore, the integrated approach to the analysis of bodies proposed in this manuscript is adequate. Only there is no mention of prostheses that allow you to identify a deceased person.

In general, the article systematizes various aspects of the study of corpses with varying degrees of alteration. After a minor revision of the manuscript, I recommend accepting it for publication in the Journal «Diagnostics».

I hope that my comments will be useful to the authors.

Author Response

REVIEWER 2

Dear Reviewer,

 thank you for your comment; the suggested changes have been performed and you can find the manuscript modifications as follows: the added text in red; the deleted text in blue. Tables and figures have been modified.

This manuscript presents the results of investigations on a relevant subject matter of Journal «Diagnostics». The manuscript presents an analysis of approaches to forensic examination of bodies. In my opinion, this is an extremely important problem in forensic medicine, but the manuscript needs to be improved.

1). English cannot be considered as a criterion, since murder is a crime against humanity. It is necessary to use all available references so as not to miss important information. In addition, excellent text translators are available on the Internet.

Thank you. We have extended the review to include articles written also in other languages than English.

2). Tables 1 and 2: Remove table break in the middle of the page (Table 1(2). (Continued)).

Thank you for your report. We have made the change in the tables.

3). The title of section 3.1 and 3.3 are the same. Change the titles to indicate the specific content of the section.

Thank you. We have changed the title of 3.3 paragraph.

4). Of course, DNA analysis is an important part of personality identification, but on its basis it is impossible to establish the cause and date of death. Therefore, the integrated approach to the analysis of bodies proposed in this manuscript is adequate. Only there is no mention of prostheses that allow you to identify a deceased person.

Thank you for the suggestion. The use of prostheses for identification purposes has been discussed.

Round 2

Reviewer 1 Report

congratulations!